# Body Image and the Double Burden of Nutrition among South Africans from Diverse Sociodemographic Backgrounds: SANHANES-1

**DOI:** 10.3390/ijerph17030887

**Published:** 2020-01-31

**Authors:** Zandile June-Rose Mchiza, Whadi-ah Parker, Ronel Sewpaul, Sunday Olawale Onagbiye, Demetre Labadarios

**Affiliations:** 1School of Public Health, University of the Western Cape, Bellville 7535, South Africa; sonagbiye@uwc.ac.za; 2Social Aspects of Public Health, Human Sciences Research Council, Cape Town 8000, South Africa; wparker@hsrc.ac.za (W.-a.P.); rsewpaul@hsrc.ac.za (R.S.); 3Faculty of Medicine and Health Sciences, Stellenbosch University, Tygerberg 7505, South Africa; dlabadarios@cybersmart.co.za

**Keywords:** body image, body mass index, socio-demography, body size dissatisfaction, body size misreporting, age, household income, ethnicity, gender, education level

## Abstract

This study investigated the associations between underweight, obesity and body image (BI) among 15+ year-old South Africans with diverse socio-demographic backgrounds. A cross-sectional survey and the analyses of data for 6411 15+ year-old participants in the first South African National Health and Nutrition Examination Survey was undertaken. Body image was compared to body mass index (BMI) and socio-demography. Data were analyzed using SPSS versions 25. Results are in percentages, means, 95% confidence intervals, p-values, and odds ratios. Overall, participants who were obese of which majority: were females, earned ZAR 9601+, completed grade 6, were non-Black men, were married and resided in urban formal areas, were more likely to underestimate their BMI and desire to be lighter. Participants who were underweight of which majority: were males, had no form of income or education, were black men, were not married, resided in less urban and farm areas, were younger than 25 years, were more likely to overestimate their BMI and desire to be heavier. While underweight and obesity were strong determinants of BI, BI was differentiated by socio-demography. These findings have a public health implication that requires special attention to curb the irrepressible underweight and obesity in South Africa.

## 1. Introduction

Underweight and obesity co-exist in the same households in South Africa [1,2]. While the prevalence of underweight in men is 4 times that of women, the prevalence of obesity in women is almost 4 times that of men [2]. Body image seems to be one of the important factors that can explain the stable underweight prevalence in men and the escalating prevalence of obesity especially in women and those individuals who fall below the poverty index line (PIL) in South Africa [1,2,3,4]. In fact, Shisana et al. [1] and Mchiza et al. [5] have shown that the majority of South Africans who present with negative body image (BI) tend to also present with unhealthy body size (underweight and obesity). Globally, there is an increasing awareness of BI [6]. In the “Western” affluent world, there are persistent pressures that promote the idea that being lighter in body size is attractive [7]. In less urbanized poorer communities, socio-cultural pressures promote the idea that being heavier in body size is attractive [7,8]. These pressures contribute to dissatisfaction about body size appearance with individuals tending to perceive their body mass index (BMI) as not normal if it is not lighter or heavier [3]. Body mass index dissatisfaction results in psychological problems that are linked to BI concerns [4].

South Africa is rapidly becoming Westernized [9]. Westernization has been implicated in the development of obesity (*BMI* > 30 kg/m^2^) and its co-morbidities in the country [9]. Concurrently, BI concerns in the country have increased unevenly [1,10,11,12] and this is thought to further influence the development of nutritional disorders [13]. There is also developing South African evidence that suggests that after age, marital status seems to be the strongest predictor for overweight, and abdominal obesity in both men and women, with this somewhat associated with BI [2]. However, the relationship between marital status, BMI and BI is barely understood. 

South Africa is recognized as a nation with diverse socio-cultural pressures brought about by diverse ethnicities, education and income levels, as well as the geographic location [14]. According to Statistics South Africa [15], more than 25 years after the end of apartheid, racial differences in the country still exist, with the unemployment rate having escalated to reach 29%. The majority of unemployed South Africans reside in peri- or non-urbanized communities and the vast majority is Black [15]. These South Africans contend with poverty and constraints related to access to treated water, sanitation and electricity. 

There is substantiated South Africa evidence which suggests that both underweight and obesity and their health co-morbidities are most noticeable in communities where poverty is high [1,2,3,12,13,16]. Furthermore, in the Wittenberg [12] study that reviewed South African small studies on BMI that also researched BI, it was observed that there was a syndemic [17] of nutrition (i.e. undernutrition and over-nutrition in the same communities). In this study, underweight individuals showed desires to be heavier. Similarly, insufficient “economic success” left a large majority of overweight South Africans also having feelings of body size lightness as their underweight counterparts. Because of this, these overweight individuals desired to be heavier than what they were. Wittenberg’s [12] research was not sufficient in explaining the nexus between the socio-demographical characteristics, BMI and BI on a representative national sample. 

The current study, therefore, attempts to bridge gaps in the research of Wittenberg [12] by explaining the relationship between underweight, obesity and BI among South Africans with diverse socio-demographic characteristics. The results of this paper will be used as a platform to develop targeted interventions to curb the nutrition syndemic and dispel the myths about body size, especially in disadvantaged populations in South Africa.

## 2. Methods and Procedures 

A detailed presentation of the methods used in the current research can be found in the article by Mchiza et al. [8]. In summary, this is a descriptive and exploratory cross-sectional survey and secondary analyses of data for 6411 South Africans (15+ years) participating in the the 1st South African National Health and Nutrition Examination Survey (SANHANES-1) [1]. 

South Africa is a multi-ethnic country with a total population of 51,770,560 based on the 2011 Census [14] that was used to select participnats for SANHANES-1. In this round of analyses, data were standardized (weighted) to represent the South African ethnic diversity, geographic location (in terms of provinces and urban/rural area of residence) and gender, based on the 2011 Census [14]. As a result, the current survey applied a multi-stage disproportionate, stratified cluster sampling approach. Enumeration areas (EAs) or groups of dwellings, within the area of research interest were used as the primary sampling units. Ten thousand households were selected from the 500 selected EAs. Of the 10,000 households, 8166 households were found to be valid, occupied households; with the rest being abandoned households. Within each household, all members were eligible to participate in the survey. The 8166 households yielded 8776 eligible individuals, 15 years or older who consented to answer questions for the 2012 SANHANES-1 BI survey. Of these 8776 individuals, only 6411 (73.1%) had valid recorded weight and height readings as well as valid calculated BMI and complete answers to the questions regarding BI, and were therefore included in the current analyses.

Participants answered questions regarding their BI and socio-demographic characteristics namely: age, gender, ethnicity, geographic location, socioeconomic status (defined as household income per annum [ZAR, representing South African Rands]), marital status and education level. These questions were outlined in an age and culturally sensitive previously validated questionnaire [7]. 

### 2.1. Anthropometric Variables

Bodyweight and height of the 6411 participants were assessed using the techniques described by Lee and Nieman [18]. Actual BMI was calculated as weight (in kilograms) divided by the square of height (in meters) for adults (19+ years) and age (in years) for children (≤18 years) and expressed as kg/m^2^ and percentiles for adults and children, respectively. The recommended Centers for Disease Control (CDC) BMI-for-age (indicated as a percentile) [19] and BMI (indicated as kg/m^2^) [20] cut-off points for children (15–18 year-olds) and adults (19+ year-olds), respectively were used. Underweight was defined as *BMI* ˂ 5th percentile and *BMI* ˂ 18.5 kg/m^2^ for children and adults, respectively. Normal weight was defined as *BMI* = 5th–84.9th percentile as well as *BMI* = 18.5 kg/m^2^–24.9 kg/m^2^ for children and adults, respectively. Overweight was defined as *BMI* = 85th–94.9th percentile; and *BMI* = 25 kg/m^2^–29.9 kg/m^2^ for children and adults, respectively. Finally, obesity was defined as *BMI* ≥ 95th percentile and *BMI* ≥ 30 kg/m^2^ for children and adults, respectively [19,20]. 

### 2.2. Body Image Measurements

Body image was explored in terms of both body size perception (with the negative outcome regarded as distortion) and dissatisfaction, and these measurements were based on the nine silhouettes developed by Stunkard et al. [21]. 

The perception which is presented as body image distortion (BID) index scores were calculated by subtracting the scores selected by the participants to estimate their BMI (“feel” silhouette), from the scores allocated by the researcher as the participants’ actual BMI based on the measurements undertaken at the time of the survey. The silhouettes were also allocated a BMI measurement score based on the categories outlined in Bulik et al. [22].

The body image dissatisfaction on the other hand utilized the feel–ideal difference (FID) index scores which were calculated by subtracting the number of the silhouette selected by each participants to represent their “ideal silhouette” body size from the number of the silhouette selected by the participants to estimate their BMI at the time of the study (“feel silhouette”). 

The calculated FID index scores that were equal to zero indicated contentment about body size. The scores that were further from zero (those that fell on the extremes sides, i.e. below or above zero) indicated greater body size dissatisfaction. Whereas, those scores that were closer to zero indicated less body size dissatisfaction with BMI status. 

Similarly, the BID index scores that were equal to zero were indicative of the ability to correctly estimate one’s BMI. Whereas, the scores further from zero, either on the negative or the positive side, indicated the inability to estimate BMI correctly. Finally, the scores that were closer to zero were indicative of close to correct estimation of BMI. Negative scores for BID indicated overestimation of BMI. Positive scores for BID, on the other hand, indicated underestimation of BMI.

### 2.3. Data Analyses

Data analyses were undertaken using SPSS version 25 Armonk, NY, USA, and Microsoft Excel 2016 Danvers, MA, USA. In the current analyses, the survey sample was stratified by province and enumerator areas. In this regard, “svy” was used to account for unequal sampling probabilities in order to benchmark (standardize) the sample to represent the South African Census 2011 [14] population estimates. Weighted data were analyzed using univariate and bivariate analysis techniques. Mean BMI and BMI percentiles were used to demonstrate the prevalence of underweight, normal weight, overweight and obesity. The mean BID and FID index scores were also reported based on the BMI. Estimates in counts (numbers), prevalence rates (percentages) and means were reported including prevalence corresponding to 95% CIs and standard error of means. Any differences in CI values were considered to be significant if they did not overlap. This was further confirmed using the p-values that were below 0.05. Bonferroni’s post hoc analysis test was used to evaluate between-group significant differences. Binary and multinomial logistic regression analyses were undertaken to predict the contribution of each socio-demographic characteristic to BI. Income was the variable with highest proportion of missing data. This was due to the inclusion of participants who were unable to estimate their household income. These included those participants below the working age (i.e., adolescents) and those that were unemployed. As such, data analyses including income had to be further restricted to 50% and 53% of the main population (*N* = 1140 and *N* = 2218 for men and women, respectfully) and also analyzed separately from the other data. This was then categorized under 3 outcomes (no income, low income ≤ 38,400 South African Rands (ZAR—United States Dollars (USD) to South African Rands (ZAR) = 1 USD equal to 14.41 ZAR) [suggesting income that is not enough to buy a healthy basket of food that offers enough energy and nutrients for the entire household per year] and medium to high household income > 38,400 ZAR [suggesting income that is enough to buy a healthy basket of food that offers enough energy and nutrients for the entire household per year]). Furthermore, to assess other socio-demographic characteristics, participants were given options to choose from 2 categories (male or female) for gender, 4 categories ([Black, White, Mixed Ancestry or Indian] for ethnicity, [married, cohabiting/living together, never married or divorced/separated/widowed] for marital status and [urban formal, urban informal, rural formal and rural informal] for geographic location, 5 categories (no education or achieved grades 1 to 7 or grades 8 to 11 or grade 12 or tertiary education) for education level, and 7 categories (15–18, 19–24, 25–34, 35–44, 45–54, 55–64 and 65+ years) for age. Breaking these variables into many categories was mandatory to allow successful analyses, given the sensitivity of the BI and health issues within the afore-mentioned different categories in South Africa [1,2,8,12,15]. Odds Ratios (ORs) were used as measures of association, with *p*-values that were less than 0.05, 0.01 and 0.001 used to indicate statistical significance.

### 2.4. Ethical Standards Disclosure

The SANHANES-1 Survey received ethics approval from the Research Ethics Committee of the Human Science Research Council (HSRC) of South Africa (REF: REC6/16/11/11). All participants signed written informed consent forms which were explained to them. Children below the age of 18 received consent from their parents or guardians and were also given a chance to agree to participate. 

## 3. Results

### 3.1. Socio-Demographic Profile and Anthropometric Status 

Socio-demographic and BMI details of the 6411 participants identified for inclusion in this study are outlined elsewhere [2,8]. In summary (Appendix A), the majority of the participants were: females (57.6%), 19–24 years (16.5%), Black South Africans (82.4%), residents of urban formal settings (51.1%), lived in the Gauteng province (24.7%), never married (45.8%), completed grades 8–11 (39.1%), did not have income (34.0%) or had a household income of between ZAR 9601 and ZAR 38,400 (38.6%) per annum. 

Overall, 7.0%, 42.1%, 23.0% and 27.8% of the participants were underweight, within the normal BMI range, overweight and obese, respectively. The overall mean BMI was 26.8 kg/m^2^, with female and male mean BMI within the overweight (28.9 kg/m^2^) and normal weight (23.8 kg/m^2^) ranges, respectively. Nearly one-third (32.5%) of the males were overweight and obese compared to twice as many (64.3%) females. By contrast, 11.6% and 3.7% of males and females were underweight, respectively. Less than a quarter (20.9%) of the youngest age group (15–18 year-olds) were overweight and obese with 14.0% underweight, while more than 30.0% of participants in all other age groups were overweight and obese and less than 10.0% of them were underweight. 

Overweight and obesity were the highest in urban formal settings (55.0%) compared to all other settings (range: 41.5%–48.0%). The North West (regarded as one of the rural provinces) [14] was the only province in which the combined prevalence of overweight and obesity was less than 40%, while about half (45.8%–55.8%) of the participants in all the other provinces were overweight and obese. More than two-thirds of participants who were married (64.7%) or divorced/separated/widowed (68.0%) were overweight and obese, compared to 52.4% and 40.6% of participants who lived with partners and those who were never married, respectively. More than two thirds (67.4%) of participants who had completed higher education were overweight and obese compared to about half (47.4%–53.5%) of all other participants. Close to half (45.2%) of the participants in the no-income group were overweight and obese, whereas more than two thirds (68.1%) of participants whose household income per annum was between ZAR 38,401–ZAR 153,600 and a quarter (74.8%) of those who earned more than ZAR 153,600 per annum, were overweight and obese.

### 3.2. The Direction of Body Image Dissatisfaction (FID) and Body Image Descripancy (BID) in Relation to Demographic Characteristics

Overall, South Africans seemed to underestimate their BMI (obtained a positive mean BID index score of 0.21 (Table 1)) and desired to gain weight (obtained a negative mean FID index score of −0.08 (Table 2)). However, there was a distinct significant difference between “under- and normal weight” as well as “overweight and obese” South Africans. In this case, those participants who were “under- and normal weight” seemed to overestimate their BMI and desired to gain weight (obtained negative scores for both FID and BID); while those who were “overweight and obese” seemed to underestimate their BMI and desired to lose weight (obtained positive scores for both FID and BID). Under- and over-estimation of BMI and aspirations to lose or gain weight were the highest at the extreme ends of the BMI spectrum (i.e., in those that were underweight and obese).

These distinct significant differences for both FID and BID index scores had also been shown between males and females (Table 1 and Table 2). While males achieved negative scores, females achieved positive scores. This implies that while males generally overestimated their BMI, they also indicated desires to be heavier than what they perceived their BMIs to be. The converse was shown for females in that they tended to underestimate their BMI while concurrently expressing desires to be lighter than what they perceived their BMI to be. Regardless of gender, those participants who were “under- and normal weight” seemed to overestimate their BMI and desired to gain weight, while those who were “overweight and obese” seemed to underestimate their BMI and desired to lose weight. In these cases, the only noticeable difference (that was also significant) was that overestimation of BMI in the normal weight females was somewhat less than that of their male counterparts (BID index scores: −0.21 vs. −1.36 (Table 2)).

When it comes to age, while 45–64 year-old participants desired to be lighter than what they perceived their BMI to be (obtained positive FID index scores), 15–44 and 65+ year-old participants desired to be heavier than what they perceived their BMI to be (obtained negative FID index scores) (Table 1). In this case, the only significant differences observed for FID were between the youngest age group (15–18 years) and the 25+ year-olds as well as between ages 15–34 year-olds and the 45–64 year-olds. When it comes to BID (Table 2), the index scores were only negative for 19–24 year-olds, implying that they overestimated their BMI. This was significantly different from the BID index scores of all the other age groups, who seemingly underestimated their BMI (as denoted by their positive BID index scores). Moreover, regardless of age, those participants who were “under- and normal weight” seemed to overestimate their BMI and desired to gain weight, while those who were “overweight and obese” seemed to underestimate their BMI and desired to lose weight. The only exceptions observed were with the: 15-18 year old normal weight group, 25–34 year and 65+ year old overweight groups that underestimated their BMI (obtained a positive BID index score of 0.07), and desired to be heavier than what they perceived their BMI to be (obtained negative FID index scores of = −0.03 and −0.08), respectively. 

According to ethnicity, while Black South African participants had negative FID index scores that were also significantly different from the positive FID index scores of other ethnic groups (Table 1), Indian South African participants had negative BID index scores (Table 2), but these were not significantly different from the positive BID index scores of the other ethnic groups. This indicated that Black South African participants desired to be heavier than what they perceived their weights to be, while participants from other ethnic groups desired to be lighter than what they perceived their weights to be. Indian South Africans on the other hand, overestimated their BMI, while participants from other ethnic groups underestimated it. Moreover, regardless of ethnicity, those participants who were “under- and normal weight” seemed to overestimate their BMI and desired to gain weight; with the exception of normal weight White and Indian South African participants. These participants desired to be lighter than what they perceived their weights to be (they obtained positive FID index scores of 0.43 and 0.17, respectively) (Table 1). On the other hand, they overestimated their BMI (they obtained positive BID index scores) (Table 2). It is important to note that in this case, the aspirations to lose weight were higher in the White South African participants than in the Indian South African participants (FID index score was = 0.43 in the White South African group when compared to 0.17 of the Indian South African group). While overweight and obese participants in all ethnic groups desired to lose weight and underestimated their BMI (i.e., obtained positive FID and BID index scores), overweight Black South Africans showed a tendency to desire to gain weight, although this was to a lesser extent (they obtained an average FID index score of −0.02).

Participants residing in urban formal settings achieved positive FID and BID index scores, while those in all other geographic locations achieved negative FID index scores and those residing in farms achieved negative index scores for BID (Table 1 and Table 2). The FID index scores were significantly different between urban formal residents compared to residents from other geographic areas (Table 1). This implies that, while urban formal residents desired to be lighter than what they perceived their BMI to be; participants from other geographic locations desired to be heavier than what they perceived their BMI to be. The BID index scores, however, were significantly different between the residents of farms compared to the residents of other geographic locations (Table 2). This implies that rural formal residents overestimated their BMI, while those in other geographic locations tended to underestimate their BMI. Moreover, regardless of geographic location, those participants who were “under- and normal weight” seemed to overestimate their BMI and desired to gain weight. The exception was with overweight residents from urban and rural informal areas who expressed desires to gain weight (as shown by their negative FID index scores of −0.06 and −0.25, respectively that were also significantly different from the FID scores of residents of other geographic locations) (Table 1). 

Significant differences between urban and rural areas were re-enforced by results from provincial data comparisons (data not shown). Predominantly rural provinces [14] such as the North West and Eastern Cape had significantly different FID index scores compared to the more urban provinces such as the Gauteng, and Western Cape. While rural residents tended to express desires to be heavier, residents in the most urbanized provinces, (Western Cape and Gauteng) expressed a desire to be lighter. Furthermore, participants in the North West province achieved negative and significantly different mean BID index scores compared to the scores for Gauteng and the Western Cape provinces, which were positive; thereby indicating that rural residents overestimated their BMI while urban residents underestimated their BMI. 

Participants who had never been married achieved a significantly different and negative mean index score for FID (Table 1) compared to the other marital status groups, who achieved positive scores for FID. This indicated that they desired to be heavier than what they perceived their BMI to be, while their marital status counterparts desired to be lighter than what they perceived their BMI to be. In terms of BID (Table 2), all the participants regardless of their marital status achieved the positive scores. The only significant differences observed in this case were between those who were never married and those who were either married, widowed, separated or divorced. Moreover, regardless of marital status, those participants who were “under- and normal weight” seemed to overestimate their BMI and desired to gain weight. The exception was with overweight participants who were cohabiting (living together) and those who were widowed, separated or divorced who expressed desires to gain weight (as shown by their negative FID index scores of −0.03 and −0.15 (Table 1)), hence they underestimated their BMI. The only significant differences observed in this case were between FID index scores of overweight and obese participants.

With regard to education level, participants who had not completed matric (i.e., completed grade 11 or lower), achieved negative FID index scores, while those who completed matric (grade 12) and/or furthered their education, achieved positive scores (data not shown). Significant differences were observed between these groups. This implied that participants who completed grade 11 or lower desired to be heavier than what they perceived their BMI to be, while those who completed matric (grade 12) and/or furthered their education desired to be lighter than what they perceived their BMI to be. With regard to BID index scores, while participants with primary education or less could correctly estimate their BMI, those who had secondary education or higher underestimated it. No significant differences were observed in this case. Moreover, regardless of their education level, those participants who were “under- and normal weight” seemed to overestimate their BMI and desired to gain weight. The exception was with overweight participants who had primary education or less who tended to correctly estimate their BMI as indicated by their BID index score of 0.00. This score was significantly different from the scores of underweight, normal weight and obese participants in this marital status group.

Finally, participants who reported having household income of *ZAR* > 38,400 per annum had positive mean FID index scores that were also significantly different from those with a household income of *ZAR* ≤ 38,400 per annum who had negative scores (Table 1). This implied that participants who had household income of *ZAR* ≤ 38,400 per annum desired to be heavier than what they perceived their BMI to be, while those who had household income of *ZAR* ≥ 38,400 per annum desired to be lighter than what they perceived their BMI to be. On the other hand, participants who reported having no form of income achieved a mean BID index score of 0.00 indicating that they could correctly estimate their BMI (Table 2). Participants with a household income that was equal to *ZAR* > 0.00 per annum also achieved positive BID index scores that indicated underestimation of BMI. Moreover, regardless of their income level, those participants who were “under- and normal weight” seemed to overestimate their BMI and desired to gain weight. The exception was with overweight participants who had household income of *ZAR* ≤ 38,400 per annum who obtained negative FID index scores of −0.03 and −0.07, respectively that were also significantly different from the scores of underweight, normal weight and obese participants in this marital status group. This indicated that they desired to be heavier than what they perceived their BMI to be.

It has to be noted that all significant relationships indicated by CIs were also confirmed using the Chi-squared tests for overall associations and Bonferroni significance tests within-group associations. In this case, all the *p*-values proved to be < 0.001. 

### 3.3. Age−Adjusted and Fully Adjusted Multinomial Logistic Regression Analysis of the Determinants of FID and BID Index Scores 

When data were adjusted for age in seven categories (15–18, 19–24, 25–34, 35–44, 45–54, 55–64 and 65+ years) (i.e., Table 3, Model 1), we observed evidence that: being male, having no form of schooling, being unmarried and having less or no household income (i.e., income *ZAR* ≤ 38,401) were associated with an increased likelihood of overestimating body size and a less likelihood of underestimating body size in this group of participants. These likelihoods remained even after the data were fully adjusted for age and all other sociodemographic variables included in the current analysis (i.e., Table 3, Model 2). Being Black, White and Mixed Ancestry South African, on the other hand, was associated with an increased likelihood of underestimating body size (Table 3, Model 1). Moreover, being White was associated with a less likelihood of overestimating body size in this group of participants (Table 3, Model 2). These likelihoods remained even after the data were fully adjusted for age and all other sociodemographic variables included in this analysis (Fully adjusted, Table 3, Model 2). The geographic location seemed not to show any association with BID.

Similarly, when data were adjusted for age in seven categories (15–18, 19–24, 25–34, 35–44, 45–54, 55–64 and 65+ years) (i.e., Table 4, Model 1), we observed evidence that: being male, having matric education or less, being unmarried and having less or no household income (i.e., income *ZAR* < 38,401) were associated with an increased likelihood of desiring to be heavier in body size and a less likelihood of desiring to be lighter in body size in this group of participants. These likelihoods remained even after the data was fully adjusted for age and all other sociodemographic variables included in the current analysis (Fully adjusted, Table 4, Model 2). Living in urban areas, on the other hand, increased the likelihood of desiring a lighter body size by 2 times or higher, whereas the likelihood of desiring a heavier body size was lower (Table 4, Model 1). These likelihoods remained even after the data were fully adjusted for age and all other sociodemographic variables included in Table 4 (Fully adjusted, Table 4, Model 2).

Finally, there were no changes in BID and FID OR outcomes when data were adjusted for gender, ethnicity, income and marital status separately. These outcomes may be viewed from Appendix A.

## 4. Discussion

The current analyses attempted to explain for the first time under and over-nutrition in relation to BI among South Africans aged 15 years and older with diverse socioeconomic backgrounds. Notable findings were that overall, South Africans at the extreme ends of the BMI spectrum (i.e., those who were underweight and obese) struggled to correctly estimate their body size and were highly dissatisfied with what they perceived their BMI to be. Despite the fact that South Africa is rapidly becoming Westernized [9], nutrition syndemic still exists (overweight and obesity exist in the same populations). Moreover, South African citizenry seem to show mixed views about body size heaviness and lightness [6,7,8,10,11], and these views resemble the views of people residing in low income and mostly rural countries as well as developed and mostly Westernized countries. What is even interesting is that socio-demography predicts BI views among different population groups of South Africans. For instance, according to the regression analyses carried out in the current research, gender, income, age and ethnicity seemed to be the strong predictors of BI. Even though other factors such as the education level, geographic location and marital status showed less effects on BID and FID index scores, they cannot be underestimated in the body size epidemiology in South Africa. 

According to international literature, BI is shown to influence the etiology of obesity [13,23]. However, in South Africa it is difficult to comprehend BI due to the fact that South Africa is a nation of diversity in terms of socio-demographic-characteristics [14,15]. While some South African studies have shown the BI theory that Black, poorer and rural individuals accentuate body size heaviness and regard it as a symbol of beauty, autonomy, economic freedom, health and fertility [10,13,16,24,25,26,27,28]. A different BI theory exists in the White, affluent and urban South African communities, where body size lightness is preferred and excessive weight gain is denigrated [10,11,28]. It has to be noted that these theories have been proven using small and localized population study samples while also utilizing qualitative research methodology. The current research therefore, employed a quantitative and exploratory research methodology to explore these theories in a representative national South African sample. In fact, in the current research, South Africans from diverse backgrounds were fully represented, and the BI information obtained from these groups could be successfully compared. 

From this study it also appears as though South Africans who perceive, or have been told that they are underweight or obese have internalized these social stereotypes and as such, this affected their psychological state. Hence, they seem to feel bad when probed about their body size status, and as such they misreport it by telling the probing person that they were highly dissatisfied by it and desired to make body size changes. These situations have been previously shown in similar body size research undertaken internationally [29,30]. 

The findings of the current research also show that in South Africa there seem to be strong relationships between BMI, BI and socio-demographic characteristics. In fact, in this analyses, the mean BID and FID index scores appeared to be negative (indicating an overestimation of body size and increased the desire to be heavier) for the most part in groups who had socio-demographic characteristics that are viewed as “inferior” in the South African context. These characteristics include young age, experiencing financial constraints, being uneducated or having less education, being Black, as well as living in rural or under resources areas [15]. On the other hand, in this analysis, the mean BID and FID index scores appeared to be positive (indicating underestimation of body size and increased desires to be lighter) for the most part in those groups who have socio-demographic characteristics that are viewed as “superior” in the South African context. These characteristics include adulthood, affluence, high education level, being non-Black and living in fully resourced or urbanized formal areas [15]. However, it also has to be noted that in this analysis, most groups that were viewed as “inferior” presented with the highest prevalence of underweight while those who are viewed as “superior” presented with the highest prevalence of obesity [2]. 

What is also interesting in the current analyses is that the findings obtained seemed to corroborate those of Wittenberg [12] in that, overweight Black South Africans (women, in particular) tended to have greater aspirations to gain weight, despite the fact that they were already heavier in body sizes at the time the research was carried out. Similarly, less affluent, rural and urban informal residents who were also heavier in body size had feelings of body size lightness brought about by their constrained economic success. Because of this, they desired to be heavier than what their BMI were at the time the research was carried out. These outcomes could be explained by the fact that, in the Black, poorer and informal communities of South Africa, body size heaviness is still equated to affluence and economic freedom [12]. Moreover, in these communities, females who are heavier are regarded as healthy (i.e., are free of HIV) [26,27], beautiful or are perceived as being treated well by their husbands or partners [5]. These observations, therefore, support the fact that while the BMI plays a major role in the BI etiology, socio-demographic factors (especially gender, ethnicity and household income) mitigate this BMI and BI relationship. 

Finally, amidst all the strengths of the current research, there are limitations that need to be considered. The first one is presenting information on adults and adolescents in the same manuscript. While this issue could be viewed as the strength of the current research (since it warranted researchers to successfully do age comparisons), it could however, mask the uniqueness of factors associated with BI in adolescents. However, this could not be avoided given the nature of the sample selection design for the SANHANES-1 that is of household nature. We could not exclude adolescents who were available to be interviewed within households at the time of the research. Moreover, we only managed to interview very few of these adolescents, reasons being that most were not at home at the time of the survey (i.e., they were possibly at learning institutions). As such, it was not going to be useful to present the data of the small group of adolescents we managed to get separately. 

Another limitation is that a number of factors, including peer pressure and living with diseases such as HIV and TB that have been shown to have a strong influence on how people view their body sizes could not be included in our study. Investigating these factors was beyond the scope of this research.

## 5. Conclusions

To conclude, this research managed to magnify the nexus between BMI, BI and different socio-demographic variables in a representative sample of South Africans. Notable findings were that those South Africans who are obese or underweight appear to have highly distorted BI than normal weight and overweight South Africans. As a result, they seem to be overly concerned about their body size, hence they are highly dissatisfied with what their BMI status was. The BMI–BI relationship in this analysis in most part was mediated by the diversity in socio-demographic characteristics, and to a larger extent gender, household income and ethnicity. These outcomes, therefore, corroborate other South African and international study outcomes undertaken in small population sample groups [9,12,29,30,31,32]. These outcomes, also supplement the BI literature in South Africa. They also call for the South African government to improve economic freedom for the majority of South Africans and endorse the development of targeted interventions to advocate positive BI. Doing this will form basis for curbing the almost stable prevalence of underweight and the escalating prevalence of obesity, eating and exercise disorders and their co-morbidities in the country.

## Figures and Tables

**Table 1 ijerph-17-00887-t001:** Mean body size dissatisfaction / feel–ideal discrepancy (FID) scores of adult South Africans aged 15 years and older, SANHANES 2012 ^1^.

	Total		Underweight		Normal Weight		Overweight		Obese	
	Mean FID	CI	Mean FID	CI	Mean FID	CI	Mean FID	CI	Mean FID	CI
**Males**	−0.32	[−0.42, −0.22]	−0.9	[−1.17, −0.63]	−0.62	[−0.74, −0.50]	0.18	[−0.00, 0.37]	0.77	[0.56, 0.98]
**Females**	0.09	[0.02, 0.16]	−1.1	[−1.38, −0.81]	−0.56	[−0.66, −0.46]	0.03	[−0.08, 0.13]	0.76	[0.66, 0.87]
**Total**	−0.08	[−0.15, −0.02]	−0.96	[−1.15, −0.76]	−0.59	[−0.67, −0.51]	0.09	[−0.01, 0.19]	0.76	[0.66, 0.87]
**15–18**	−0.45	[−0.58, −0.32]	−1.15	[−1.47, −0.82]	−0.51	[−0.67, −0.34]	0.18	[−0.13, 0.48]	0.31	[−0.24, 0.85]
**19–24**	−0.26	[−0.36, −0.16]	−0.75	[−1.06, −0.44]	−0.57	[−0.69, −0.45]	0.09	[−0.09, 0.27]	0.76	[0.48, 1.05]
**25–34**	−0.11	[−0.22, −0.00]	−0.78	[−1.21, −0.35]	−0.58	[−0.74, −0.41]	−0.03	[−0.24, 0.17]	0.73	[0.51, 0.94]
**35–44**	−0.05	[−0.20, 0.11]	−1.15	[−1.47, −0.83]	−0.71	[−0.97, −0.45]	0.05	[−0.17, 0.28]	0.84	[0.62, 1.06]
**45–54**	0.17	[0.03, 0.31]	−0.6	[−1.23, 0.04]	−0.69	[−0.93, −0.46]	0.3	[0.04, 0.56]	0.83	[0.65, 1.02]
**55–64**	0.14	[−0.02, 0.31]	−0.95	[−2.02, 0.11]	−0.51	[−0.83, −0.19]	0.11	[−0.10, 0.32]	0.8	[0.62, 0.98]
**≥65**	−0.08	[−0.22, 0.06]	−1.38	[−1.86, −0.90]	−0.66	[−0.88, −0.45]	−0.08	[−0.34, 0.17]	0.63	[0.43, 0.84]
**Total**	−0.08	[0.01, −0.02]	−0.45	[0.31, −0.58]	−0.26	[0.12, −0.33]	−0.11	[0.00, 0.22]	−0.65	[0.41, 0.44]
**Black**	−0.17	[−0.25, −0.10]	−0.99	[−1.23, −0.76]	−0.67	[−0.76, −0.58]	−0.02	[−0.14, 0.09]	0.67	[0.56, 0.79]
**White**	0.73	[0.40, 1.06]	−1.53	[−2.53, −0.53]	0.43	[−0.06, 0.92]	0.59	[−0.08, 1.27]	1.1	[0.58, 1.61]
**Mixed ancestry**	0.17	[0.07, 0.28]	−0.76	[−1.04, −0.48]	−0.43	[−0.53, −0.33]	0.5	[0.27, 0.74]	1.14	[0.97, 1.31]
**Indian**	0.46	[0.11, 0.81]	−0.77	[−1.05, −0.50]	0.17	[−0.06, 0.40]	0.8	[0.47, 1.12]	1.72	[1.47, 1.96]
**Total**	−0.09	[−0.15, −0.02]	−0.95	[−1.15, −0.76]	−0.6	[−0.68, −0.52]	0.09	[−0.01, 0.19]	0.76	[0.66, 0.87]
**Urban formal**	0.13	[0.03, 0.22]	−0.8	[−1.14, −0.46]	−0.47	[−0.58, −0.35]	0.3	[0.14, 0.45]	0.92	[0.78, 1.06]
**Urban informal**	−0.18	[−0.32, −0.03]	−1.13	[−1.55, −0.70]	−0.59	[−0.79, −0.39]	−0.06	[−0.26, 0.15]	0.81	[0.50, 1.13]
**Rural informal (tribal)**	−0.39	[−0.49, −0.30]	−1.27	[−1.55, −0.99]	−0.82	[−0.97, −0.68]	−0.25	[−0.38, −0.11]	0.47	[0.31, 0.63]
**Rural formal (farms)**	−0.13	[−0.25, −0.01]	−0.49	[−0.95, −0.04]	−0.44	[−0.60, −0.28]	0.05	[−0.07, 0.18]	0.63	[0.38, 0.88]
**Total**	−0.08	[−0.15, −0.02]	−0.96	[−1.15, −0.76]	−0.59	[−0.67, −0.51]	0.09	[−0.01, 0.19]	0.76	[0.66, 0.87]
**Never married**	−0.24	[−0.32, −0.16]	−1.13	[−1.34, −0.92]	−0.6	[−0.70, −0.51]	0.1	[−0.03, 0.23]	0.73	[0.57, 0.89]
**Living together**	0.01	[−0.15, 0.17]	−0.09	[−0.81, 0.64]	−0.44	[−0.65, −0.23]	−0.03	[−0.24, 0.19]	0.9	[0.54, 1.27]
**Married**	0.11	[−0.00, 0.22]	−0.87	[−1.70, −0.04]	−0.6	[−0.79, −0.40]	0.18	[−0.00, 0.37]	0.75	[0.62, 0.88]
**Widowed/Separated/Divorced**	0.08	[−0.11, 0.26]	−1	[−1.68, −0.33]	−0.95	[−1.30, −0.60]	−0.15	[−0.41, 0.10]	0.95	[0.72, 1.19]
**Total**	−0.06	[−0.13, 0.01]	−0.99	[−1.22, −0.76]	−0.62	[−0.70, −0.53]	0.09	[−0.01, 0.19]	0.79	[0.69, 0.89]
**No schooling**	−0.18	[−0.31, −0.05]	−0.84	[−1.27, −0.41]	−0.77	[−0.95, −0.59]	0	[−0.25, 0.24]	0.65	[0.48, 0.81]
**Grades 1 to 5 (primary school)**	−0.10	[−0.24, −0.01]	−0.80	[−1.61, −0.11]	−0.70	[−0.33, −0.81]	0	[−0.00, 0.00]	0.60	[0.49, 0.66]
**Grades 6 to 11 (secondary school)**	−0.08	[−0.05, 0.61]	−1.05	[−1.37, −0.97]	−0.52	[−0.21, −0.53]	0.13	[0.04, 0.12]	0.78	[0.61, 0.10]
**Grade 12 (matric)**	0.2	[−0.15, 0.00]	−1.02	[−1.24, −0.87]	−0.50	[−0.61, −0.43]	0.10	[0.02, 0.23]	0.74	[0.64, 0.91]
**Tertiary (beyond matric)**	0.4	[0.19, 0.62]	−0.24	[−0.97, 0.49]	−0.44	[−0.76, −0.12]	0.34	[−0.06, 0.74]	1.22	[0.90, 1.54]
**Total**	−0.07	[−0.14, −0.00]	−0.95	[−1.16, −0.73]	−0.58	[−0.67, −0.50]	0.11	[−0.00, 0.22]	0.78	[0.67, 0.88]
**No income**	−0.19	[−0.30, −0.09]	−1.11	[−1.36, −0.86]	−0.65	[−0.78, −0.52]	−0.03	[−0.22, 0.16]	0.78	[0.59, 0.98]
**≤38,400 ZAR ^i^**	−0.08	[−0.18, 0.02]	−0.73	[−1.27, −0.19]	−0.71	[−0.89, −0.54]	−0.07	[−0.20, 0.06]	0.76	[0.62, 0.90]
**>38,400 ZAR ^i^**	0.4	[0.22, 0.58]	−0.95	[−2.97, 1.07]	−0.24	[−0.60, 0.12]	0.28	[−0.05, 0.60]	1.05	[0.78, 1.32]
**Total**	−0.06	[−0.13, 0.02]	−0.91	[−1.23, −0.59]	−0.64	[−0.76, −0.53]	0	[−0.10, 0.11]	0.81	[0.70, 0.92]

^1^ Of the individuals for whom valid weight and height readings were recorded, and who answered the questions on perceived and ideal body size. ^i^ Household income in South African Rands (ZAR).

**Table 2 ijerph-17-00887-t002:** Mean body image discrepancy (BID) scores of adult South Africans aged 15 years and older, SANHANES 2012 ^1^.

	Total		Underweight		Normal weight		Overweight		Obese	
	Mean BID	CI	Mean BID	CI	Mean BID	CI	Mean BID	CI	Mean BID	CI
**Males**	−0.6	[−0.74, −0.45]	−2.49	[−2.77, −2.20]	−1.36	[−1.49, −1.24]	0.9	[0.70, 1.10]	2.24	[1.93, 2.54]
**Females**	0.8	[0.71, 0.90]	−2.27	[−2.53, −2.01]	−0.21	[−0.31, −0.10]	0.98	[0.86, 1.09]	1.79	[1.67, 1.91]
**Total**	0.21	[0.11, 0.31]	−2.42	[−2.63, −2.21]	−0.86	[−0.95, −0.76]	0.95	[0.84, 1.05]	1.87	[1.76, 1.99]
**15–18**	0.41	[0.19, 0.62]	−1.83	[−2.12, −1.54]	0.07	[−0.09, 0.24]	2.48	[2.19, 2.78]	4.29	[3.61, 4.98]
**19–24**	−0.45	[−0.64, −0.25]	−2.62	[−2.89, −2.35]	−1.22	[−1.39, −1.05]	0.96	[0.67, 1.24]	1.99	[1.56, 2.42]
**25–34**	0.11	[−0.06, 0.29]	−2.4	[−2.74, −2.05]	−1.05	[−1.25, −0.86]	0.8	[0.60, 0.99]	1.86	[1.60, 2.13]
**35–44**	0.34	[0.18, 0.50]	−2.37	[−2.71, −2.03]	−0.76	[−1.02, −0.50]	0.93	[0.73, 1.13]	1.71	[1.50, 1.92]
**45–54**	0.42	[0.20, 0.63]	−3.36	[−4.11, −2.61]	−1.23	[−1.47, −0.99]	0.77	[0.50, 1.04]	1.91	[1.71, 2.10]
**55–64**	0.36	[0.05, 0.68]	−2.49	[−3.44, −1.53]	−1.28	[−1.68, −0.88]	0.65	[0.44, 0.86]	1.79	[1.58, 1.99]
**≥65**	0.43	[0.21, 0.65]	−2.35	[−2.86, −1.85]	−0.86	[−1.14, −0.58]	0.78	[0.52, 1.03]	1.69	[1.43, 1.95]
**Total**	0.23	[0.20, 0.06]	−2.45	[−2.60, −1.63]	−1.10	[0.60, 0.99]	1.12	[0.90, 1.33]	1.92	[1.66, 2.18]
**Black**	0.23	[0.12, 0.34]	−2.37	[−2.62, −2.12]	−0.86	[−0.96, −0.76]	1.00	[0.88, 1.12]	1.91	[1.77, 2.05]
**White**	0.51	[−0.17, 1.18]	−2.23	[−2.73, −1.74]	−1.22	[−2.19, −0.26]	0.79	[0.17, 1.40]	1.72	[1.25, 2.20]
**Mixed ancestry**	0.09	[−0.07, 0.25]	−2.67	[−2.93, −2.40]	−0.79	[−1.07, −0.50]	0.68	[0.39, 0.97]	1.76	[1.61, 1.92]
**Indian**	−0.36	[−1.00, 0.28]	−2.66	[−2.97, −2.34]	−0.86	[−1.18, −0.53]	0.87	[0.59, 1.15]	1.29	[1.04, 1.54]
**Total**	0.21	[0.11, 0.31]	−2.42	[−2.63, −2.21]	−0.86	[−0.95, −0.76]	0.95	[0.84, 1.06]	1.87	[1.76, 1.99]
**Urban formal**	0.34	[0.17, 0.50]	−2.44	[−2.81, −2.07]	−0.89	[−1.04, −0.74]	0.94	[0.77, 1.12]	1.94	[1.76, 2.11]
**Urban informal**	0.16	[−0.08, 0.39]	−2.03	[−2.38, −1.68]	−0.69	[−0.97, −0.41]	0.99	[0.78, 1.21]	1.68	[1.36, 2.00]
**Rural informal (tribal)**	0.16	[0.02, 0.29]	−2.3	[−2.58, −2.02]	−0.8	[−0.95, −0.64]	1	[0.85, 1.15]	1.82	[1.65, 2.00]
**Rural formal (Farms)**	−0.33	[−0.55, −0.12]	−3.08	[−3.68, −2.47]	−1.11	[−1.32, −0.89]	0.73	[0.50, 0.97]	1.82	[1.50, 2.15]
**Total**	0.21	[0.11, 0.31]	−2.42	[−2.63, −2.21]	−0.86	[−0.95, −0.76]	0.95	[0.84, 1.05]	1.87	[1.76, 1.99]
**Never married**	0.01	[−0.13, 0.15]	−2.27	[−2.50, −2.05]	−0.8	[−0.93, −0.68]	1.12	[0.90, 1.33]	1.92	[1.66, 2.18]
**Living together**	0.03	[−0.31, 0.37]	−3.61	[−4.69, −2.54]	−0.97	[−1.28, −0.65]	0.82	[0.45, 1.18]	1.75	[1.25, 2.25]
**Married**	0.53	[0.34, 0.71]	−2.73	[−3.49, −1.96]	−1.05	[−1.26, −0.83]	0.93	[0.78, 1.08]	1.89	[1.74, 2.05]
**Widowed/Separated/Divorced**	0.49	[0.29, 0.69]	−2.63	[−3.30, −1.96]	−0.69	[−1.05, −0.34]	0.46	[0.19, 0.73]	1.57	[1.36, 1.77]
**Total**	0.24	[0.13, 0.36]	−2.49	[−2.74, −2.25]	−0.87	[−0.98, −0.77]	0.94	[0.83, 1.04]	1.84	[1.71, 1.96]
**No schooling**	0	[−0.18, 0.18]	−2.82	[−3.24, −2.40]	−1.03	[−1.21, −0.84]	0.72	[0.58, 0.85]	1.67	[1.50, 1.84]
**Grades 1 to 5 (primary school)**	0.12	[−1.00, 0.28]	−2.82	[−2.19, −0.26]	−1.03	[−1.28, −0.65]	0.72	[0.85, 1.15]	1.67	[1.39, −1.05]
**Grades 6 to 11 (secondary school)**	0.29	[0.11, 0.31]	−2.11	[−1.07, −0.50]	−0.73	[−1.26, −0.83]	1.11	[0.50, 0.97]	2.01	[1.25, −0.86]
**Grade 12 (matric)**	0.30	[0.18, 0.41]	−2.11	[−2.35, −1.87]	−0.73	[−0.85, −0.60]	1.11	[0.94, 1.27]	2.01	[1.83, 2.19]
**Tertiary (beyond matric)**	0.43	[0.05, 0.81]	−2.04	[−2.93, −1.15]	−1.11	[−1.63, −0.58]	0.8	[0.41, 1.20]	1.54	[1.26, 1.81]
**Total**	0.21	[0.11, 0.32]	−2.42	[−2.65, −2.18]	−0.83	[−0.94, −0.73]	0.95	[0.83, 1.07]	1.85	[1.72, 1.97]
**No income**	0	[−0.15, 0.16]	−2.58	[−2.91, −2.25]	−0.98	[−1.16, −0.80]	0.94	[0.77, 1.11]	1.87	[1.70, 2.05]
**≤38,400 ZAR ^i^**	0.17	[0.01, 0.33]	−2.68	[−3.20, −2.16]	−0.95	[−1.10, −0.79]	0.68	[0.54, 0.82]	1.62	[1.48, 1.77]
**>38,400 ZAR ^i^**	0.52	[0.16, 0.87]	−2.12	[−3.49, −0.75]	−1.22	[−1.75, −0.69]	0.78	[0.37, 1.19]	1.72	[1.46, 1.98]
**Total**	0.16	[0.03, 0.28]	−2.62	[−2.94, −2.31]	−0.99	[−1.11, −0.87]	0.77	[0.65, 0.89]	1.71	[1.60, 1.82]

^1^ Of the individuals for whom valid weight and height readings were recorded and who answered the questions on perceived and ideal body size. ^i^ Household income in South African Rands (ZAR).

**Table 3 ijerph-17-00887-t003:** The likelihood of body image discrepancy (BID) among participants from the SANHANES-1 survey by indicators of sociodemographic factors (age-adjusted and fully adjusted odds ratios from multinomial regression).

Socio-Demographic Factors		Model 1: Age-Adjusted OR (CI)			Model 2: Fully Adjusted OR (CI)		
		BID Less than Zero (<0)	BID Equal to Zero (=0)	BID Greater than Zero (>0)	BID Less than Zero (<0)	BID Equal to Zero (= 0)	BID Greater than Zero (>0)
**Gender**	Male	3.80 (3.06–4.71)	1 (< 0.001)	0.55 (0.45–0.69)	3.86 (3.12–4.79)	1 (< 0.001)	0.56 (0.45–0.69)
	Female	1		1	1		1
**Education level**	No schooling	2.18 (1.24–3.84)		0.96 (0.58–1.58)	2.36 (1.33–4.18)		0.99 (0.60–1.64)
	Grade 1–5 (primary school)	1.93 (1.12–3.30)		1.06 (0.66–1.71)	2.12 (1.23–3.66)		1.04 (0.64–1.68)
	Grade 6–7 (junior secondary school)	1.45 (0.84–2.50)	1 (< 0.001)	1.12 (0.70–1.79)	1.58 (0.91–2.75)	1 (< 0.001)	1.09 (0.68–1.76)
	Grade 8–11 (senior secondary school)	1.14 (0.70–1.85)		1.08 (0.72–1.63)	1.21 (0.74–1.96)		1.07 (0.71–1.62)
	Grade 12 (matric)	0.87 (0.53–1.42)		0.77 (0.51–1.16)	0.86 (0.52–1.41)		0.77 (0.51–1.16)
	Tertiary (beyond matric)	1		1	1		1
**Marital status**	Never married	1.17 (0.81–1.68)		0.87 (0.63–1.20)	1.15 (0.80–1.66)		0.86 (0.62–1.20)
	Living together	0.99 (0.63–1.55)	1 (< 0.001)	0.85 (0.56–1.29)	0.96 (0.61–1.53)	1 (< 0.001)	0.83 (0.54–1.26)
	Married	0.67 (0.48–0.94)		1.13 (0.84–1.50)	0.67 (0.48–0.94)		1.08 (0.81–1.45)
	Widowed/Separated/Divorced	1		1	1		1
**Geographic location**	Urban formal	0.81 (0.59–1.11)		1.21 (0.89–1.63)	0.83 (0.61–1.13)		1.22 (0.90–1.64)
	Urban informal	0.77 (0.51–1.17)	1 (0.146)	1.03 (0.69–1.51)	0.79 (0.53–1.20)	1 (0.185)	1.01 (0.68–1.50)
	Rural formal (Tribal areas)	0.77 (0.54–1.10)		1.12 (0.79–1.57)	0.79 (0.55–1.13)		1.13 (0.80–1.59)
	Rural informal (farms)	1		1	1		1
**Ethnicity**	Black	1.15 (0.69–1.89)		2.03 (1.30–3.16)	1.12 (0.68–1.85)		2.04 (1.31–3.18)
	White	0.99 (0.43–2.31)	1 (< 0.001)	1.77 (0.89–3.52)	0.98 (0.42–2.29)	1 (< 0.001)	1.83 (0.91–3.65)
	Mixed Ancestry	1.23 (0.74–2.05)		1.37 (0.87–2.16)	1.22 (0.73–2.03)		1.37 (0.87–2.16)
	Indian/Asian	1		1	1		1
**Income category (per annum) ZAR ^i^**	No income	4.60 (1.80–11.7)		0.92 (0.50–1.70)	4.33 (1.68–11.1)		0.96 (0.52–1.78)
	≤38,400	3.70 (1.43–9.88)	1 (< 0.001)	0.71 (0.39–1.40)	3.43 (1.32–9.19)	1 (< 0.001)	0.75 (0.40–1.46)
	>38,400	1		1	1		1

^i^ ZAR: South African Rands (household income). Model 1: Adjusted for age in seven categories (15–18, 19–24, 25–34, 35–44, 45–54, 55–64 and 65+ years). Model 2: Fully adjusted. Adjusted for age and all other sociodemographic variables included in the table.

**Table 4 ijerph-17-00887-t004:** The likelihood of feel–ideal discrepancy (FID) among participants from the SANHANES-1 survey by indicators of sociodemographic factors (age-adjusted and fully adjusted odds ratios from multinomial regression).

Socio-Demographic Factors		Model 1: Age-Adjusted OR(CI)			Model 2: Fully Adjusted OR(CI)		
		FID Less than Zero (< 0)	FID Equal to Zero (= 0)	FID Greater than Zero (>0)	FID Less than Zero (< 0)	FID Equal to Zero (= 0)	FID Greater than Zero (>0)
**Gender**	Male	1.56 (1.31–1.86)	1 (< 0.001)	0.54 (0.45–0.66)	1.57 (1.32–1.874)	1 (< 0.001)	0.55 (0.45–0.67)
	Female	1		1	1		1
**Education level**	No schooling	1.02 (0.63–1.66)		0.51 (0.32–0.81)	1.04 (0.64–1.70)		0.57 (0.35–0.90)
	Grade 1–5 (primary school)	1.27 (0.80–2.02)		0.70 (0.45–1.09)	1.24 (0.78–1.98)		0.67 (0.43–1.05)
	Grade 6–7 (junior secondary school)	1.14 (0.71–1.82)	1 (0.009)	0.69 (0.44–1.07)	1.11 (0.69–1.79)	1 (0.035)	0.65 (0.42–1.01)
	Grade 8–11 (senior secondary school)	0.92 (0.61–1.41)		0.74 (0.50–1.07)	0.92 (0.60–1.40)		0.72 (0.49–1.05)
	Grade 12 (matric)	0.76 (0.49–1.17)		0.69 (0.47–1.01)	0.76 (0.49–1.18)		0.68 (0.46–1.00)
	Tertiary (beyond matric)	1		1	1		1
**Marital status**	Never married	0.80 (0.59–1.09)		0.64 (0.47–0.87)	0.79 (0.58–1.08)		0.63 (0.46–0.85)
	Living together	0.87 (0.59–1.28)	1 (0.003)	0.86 (0.58–1.27)	0.83 (0.56–1.23)	1 (0.019)	0.77 (0.52–1.14)
	Married	0.79 (0.60–1.05)		0.98 (0.75–1.28)	0.76 (0.57–1.01)		0.86 (0.66–1.14)
	Widowed/Separated/Divorced	1		1	1		1
**Geographic location**	Urban formal	1.28 (0.98–1.67)		1.99 (1.49–2.64)	1.28 (0.98–1.68)		2.06 (1.55–2.75)
	Urban informal	1.57 (1.11–2.22)	1 (< 0.001)	2.31 (1.60–3.36)	1.55 (1.10–2.19)	1 (< 0.001)	2.30 (1.58–3.34)
	Rural formal (Tribal areas)	1.76 (1.31–2.36)		1.57 (1.13–2.19)	1.78 (1.33–2.39)		1.66 (1.19–2.31)
	Rural informal (farms)	1		1	1		1
**Ethnicity**	Black	1.52 (0.90–2.59)		0.70 (0.46–1.05)	1.52 (0.89–2.58)		0.69 (0.46–1.04)
	White	0.77 (0.31–1.92)	1 (0.001)	0.96 (0.51–1.79)	0.81 (0.33–2.01)	1 (< 0.001)	1.05 (0.56–1.96)
	Mixed Ancestry	1.42 (0.83–2.45)		0.99 (0.65–1.50)	1.41 (0.82–2.43)		0.98 (0.64–1.48)
	Indian/Asian	1		1	1		1
**Income category (per annum) ZAR ^i^**	No income	1.30 (0.59–2.88)		0.42 (0.23–0.77)	1.33 (0.60–2.95)		0.46 (0.25–0.85)
	≤38,400	1.34 (0.60–3.06)	1 (0.002)	0.40 (0.23–0.81)	1.35 (0.61–3.12)	1 (0.006)	0.44 (0.25–0.89)
	>38,401	1		1	1		1

^i^ ZAR: South African Rands (household income). Model 1: Adjusted for age in seven categories (15–18, 19–24, 25–34, 35–44, 45–54, 55–64 and 65+ years). Model 2: Fully adjusted. Adjusted for age and all other sociodemographic variables included in the table.

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
