# Peer review of "Body Image and the Double Burden of Nutrition among South Africans from Diverse Sociodemographic Backgrounds: SANHANES-1"

_ijerph, 2020, doi:10.3390/ijerph17030887_

Round 1

Reviewer 1 Report

Manuscript ID: ID: ijerph-677996

Body image and the double burden of nutrition among 3 South Africans from diverse sociodemographic 4 backgrounds: SANHANES-1

line 35: [1,2[. - correct [1,2] 

line 45: kgm2 - correct kg/m2

line 58: [1,2,8,,12,13,16] - correct [1,2,8,12,13,13]

line 77: [defined as household income per annum] -correct (defined as household income per annum)

line 90, 91: kg/m2 - correct kg/m2

line 140: In summary (Supplement 1)  - I don't understand what supplement. Similarly, the line 193, 214, 232,262, 316, 320, 341

line 147: 23.8kg/m2 – correct 23.8 kg/m2 (space)

line 204: obtained –ve FID index scores - I don't understand

Figure 9: <=R38400 - I don't understand, similarly, in Table 4, the bottom line <=153601

Numbers in Figures 1 – 9 - with comma or dot?

Please go thoroughly throughout the guide for authors of International Journal of Environmental Research and Public Health and make necessary technical corrections to your manuscript, e.g. font face, font size, spacing of paragraphs, gaps, dots, commas, etc.).

Conclusion: Accept after minor revision.

Author Response

Dear Reviewer,

We highly appreciate that you have taken time to supply comments to our manuscript. These surely improved it a lot.

Please receive the point-to-point corrections we have undertaken below:

 Misprints on lines 35, 49, 63, 64, 94, 95, 107 and 108 are corrected

Space on line 179 has been added Brackets on lines 94 and 95 are corrected The statements referring to Supplement 2 (i.e. lines 140, 193, 214, 232, 262, 316, 320 and 341) have been removed, as all the data referred to is now presented in presented in Tables 1 and 2. The symbol “–ve” on lines 221 and 314 is now written correctly as “negative” The symbols “<=” in Tables 3 and 4 are now written correctly as “≤” Spell check has be done for the entire manuscript Misprints on lines 35, 49, 63, 64, 94, 95, 107 and 108 are corrected Space on line 179 is added Brackets on lines 94 and 95 are corrected The statements referring to Supplement 2 (i.e. lines 140, 193, 214, 232, 262, 316, 320 and 341) have been removed, as all the data referred to is now presented in presented in Tables 1 and 2. The symbol “–ve” on lines 221 and 314 is now written correctly as “negative” The symbols “<=” in Tables 3 and 4 are now written correctly as “≤” Spell check has be done for the entire manuscript

These are also highlighted in red font in our revised manuscript.

Kind regards

..............................................

Reviewer 2 Report

The research is very interesting in that it used the first NHANES data in South Africa, which provides a nationally representative sample with weights, to examine the associations between body image, BMI, and sociodemographic characteristics. It is also interesting because of the unique context to study body image, South Africa, thus important.

Though meaningful, the manuscript is somewhat descriptive and exploratory, and the analysis needs to be reorganized.

1) There is no explanation about how socioeconomic characteristics, such as education, income, and geographic locations, were defined for analysis. For example, there is no explanation about how to interpret ZAR or how “urban” and “rural” were defined. Household income is categorized into 5 groups, but there is no justification for the cutoff points. It is unclear if grade 0-5 can be referred to elementary school or not.

2) On page 3, it is stated that a separate analysis was conducted for missing data in household income variable, but it is not well explained with one citation. If any method handling missing data, such as multiple imputations (MI), full information maximum likelihood (FIML), or hot deck, was used for the analysis, the author(s) should explain how it was done with details.

3) There is a lack of discussion about why body image matters and how it is associated with socioeconomic characteristics in South Africa in the introduction. As a result, the first part of the analysis seems rather descriptive than informative. For example, it seems that the author(s) acknowledges the importance of gender differences in body image, but the analysis was only done by weight status (figure 1), age groups (figure 3), race (figure 4), and geographical location (figure 5), provinces (figure 6), and marital status (figure 7) without considering gender. Further, for instance, the author(s) failed to provide any convincing and scientific reason to look at body image by marital status regardless of gender. In my opinion, all figures can be summarized in one table with numbers, and the second part of analysis, logistic regression results, would provide the main results.

4) On page 4, it is stated that South Africans desired to gain weight based on the fact that FID=-.08, but the 95% CI of the FID seems to include zero, which indicates non-significance. Please double check.

5) Logistic regression analysis was conducted in the second part of the analysis. There are two models with FID and BID as the outcome for each model. However, those two models have different variables in the model. For example, geographic location is considered for FID (Table 1), but not for BID (Table 2). Please provide the full information or provide a reasonable reason for exclusion. By the way, there is a typo in Table 2 (Higher education).

6) In Table 3 and 4, the term, “minimally adjusted”, was used, but it is not a common term to use when it comes to reporting statistical results. Also, the titles of Table 3 and 4 indicate that relative risk was estimated, but in the tables, odds ratios were reported. I am not sure if the author(s) intended to report relative risk or odds ratio. They are different.

Author Response

Dear Reviewer,

We appreciate the fact that you have taken time to review our manuscript and supply constructive comments.

Please see our point  - to point responses outlined below:

1)There is no explanation about how socioeconomic characteristics, such as education, income, and geographic locations, were defined for analysis. For example, there is no explanation about how to interpret ZAR or how "urban" and "rural" were defined. Household income is categorized into 5 groups, but there is no justification for the cutoff points. It is unclear if grade 0-5 can be referred to elementary school or not.

Response: Please see page 2 lines 58 to 71 and page 4 lines 144 to 161 for the full description and the justification of including the sociodemographic factors in the current analysis and the allocation of categories for each of the socio-demographic variables. Please also note that the income categories have been fused to outline 3 categories to further justify their importance and make the regression analysis successful. Further definition of education categories are outlined in Tables 1 to 4. The term “ZAR” is now defined (see line 149 and the footnote of page 4).

2) On page 3, it is stated that a separate analysis was conducted for missing data in household income variable, but it is not well explained with one citation. If any method handling missing data, such as multiple imputations (Ml), full information maximum likelihood (FIML), or hot deck, was used for the analysis, the author(s) should explain how it was done with details.

Response: the above methodologies were not undertaken since the data had to be recoded to accommodate overall household income, with each individual from the same household allocated income outcome from that particular household. Please see page 4 lines 144 to 153 for the full description.

3) There is a lack of discussion about why body image matters and how it is associated with socioeconomic characteristics in South Africa in the introduction. As a result, the first part of the analysis seems rather descriptive than informative.

Response: some information have been added in the introduction to explain why BI matters and how it associates with socioeconomic characteristics. But we highlighted that the available literature is scarce and unclear hence, we conducted the current research. Please see page 1 lines 37 to 41 and page 2 lines 51 to 54, and 58 to 71.

4) For example, it seems that the author(s) acknowledges the importance of gender differences in body image, but the analysis was only done by weight status (figure 1 ), age groups (figure 3), race (figure 4 ), and geographical location (figure 5), provinces (figure 6), and marital status (figure 7) without considering gender. Further, for instance, the author(s) failed to provide any convincing and scientific reason to look at body image by marital status regardless of gender.

Response: We do not understand comment number 3 above. In our Figure 2 we presented the outcomes of the analysis of BI by gender and also showed the outcome of BI regardless of gender in Supplement 2 Figure 1. Perhaps, the manuscript version you reviewed omitted Figure 2 and its accompanying narrative. Please note that you can now find the outcome of the analyses by gender in Tables 1 and 2 as you have suggested that the data in the figures could be successfully presented in a table format.

5) In my opinion, all figures can be summarized in one table with numbers, and the second part of analysis, logistic regression results, would provide the main results.

Response: As suggested, we now have presented all the data initially presented in Figures 1 to 9 in Tables 1 and 2. Please see the gender outcomes presented in red font.

In addition, we conducted regression analysis where we adjusted the outcomes by gender, income, marital status and ethnicity separately (Please see Supplements 2 to 4). In this case, there were no Odds Ratio differences observed. The likelihood outcomes we initially observed when adjusting by age and by all the sociodemographic variables remained.

6) On page 4, it is stated that South Africans desired to gain weight based on the fact that FID=-.08, but the 95% Cl of the FID seems to include zero, which indicates non­significance. Please double check. :

Response: The confidence intervals have been checked, thank you. It seemed as though a mistake was made, as the CI data do not include zero (-0.15 - -0.02). Please see Table 1.

7) Logistic regression analysis was conducted in the second part of the analysis. There are two models with FID and BID as the outcome for each model. However, those two models have different variables in the model. For example, geographic location is considered for FID (Table 1), but not for BID (Table 2). Please provide the full information or provide a reasonable reason for exclusion. By the way, there is a typo in Table 2 (Higher education).

Response: Some of the overall variables were dropped in the analysis because of collinearity. In fact, most of our categorical data resulted to a high correlation or an association between two or more of our potential predictor variables, hence we had to drop some of the redundant variables. This explained why only few covariates showed up in the results in our final tables. We however, removed these tables to reduce the size of this manuscript, based on the advice by reviewer 2. Since the data presented in these tables did not provide different information to that presented in Tables 3 and 4.

8) In Table 3 and 4, the term, "minimally adjusted", was used, but it is not a common term to use when it comes to reporting statistical results. Also, the titles of Table 3 and 4 indicate that relative risk was estimated, but in the tables, odds ratios were reported. I am not sure if the author(s) intended to report relative risk or odds ratio. They are different.

Response: The term "minimally adjusted" has been removed and the terms “relative risk” have been changed to the terms “likelihood” throughout the manuscript.

Finally, the conclusion has been revised

These responses are also highlighted in red font in our revised manuscript.

Kind regards

..............................................

Reviewer 3 Report

This is a very comprehensive manuscript.  There are too much information in the manuscripts so the readers may lose the track from reading it. Maybe it can be broken down to another manuscript.  Or they can have a subgroup future research in the conclusion. The conclusion and results of this manuscript are reassembled to other research that people who are obese or underweight appear to have highly distorted body image than normal weight and overweight people. The statistic that authors used were appropriate to the cross-sectional study. In the figures, it should be decimal dot not comma.  When there is a significant difference between groups, there should be symbols on the figures although they were described in the results. Please explain what ZAR is. One thing the research can do is to include the population size of South Africa since the size of the population may affect the results.

Author Response

Dear Reviewer,

We appreciate the fact that you have taken time to review our manuscript and supply constructive comments.

Please see our point  - to point responses outlined below:

The manuscript has been reduced by removing the logistic regression outcomes initially presented in Tables 1 & 2 and also presenting the data that was initially presented in Figures 1 to 9 & in the Supplement 2 in Table format (see Tables 1 and 2 in the current manuscript version). The comments relating to the Figures: we have removed all the figures and replaced them with Tables 1 and 2 based on the comments by the reviewer 2 The term ZAR is explained (see page 3 line 95) and the footnote of page 4 The sample size, participant selection and response rate are included on pages 2 and 3 (lines 81 to 92). The conclusion of the manuscript have been revised to highlight the relationship between body image, BMI and the socio-demography in South Africa.

Some of these responses are also highlighted in red font in the revised manuscript.

Kind regards

.................................................................. 

Round 2

Reviewer 2 Report

1) I wonder why the analysis was not conducted by gender. That's what I tried to point out in one of my comments, but it seems like the author(s) didn't catch that. Gender is an important piece in body image study, and body image of males and females is affected by very similar but sometimes very different factors, so it would be great if the analysis can be separated by gender as the sample size allows.

2) In Table 1, CIs for "No schooling" and "Grades 1 to 5 (primary school)" are the same. Please make it sure that they are correct.

Author Response

Reviewer 2 comments:

I wonder why the analysis was not conducted by gender. That's what I tried to point out in one of my comments, but it seems like the author(s) didn't catch that. Gender is an important piece in body image study, and body image of males and females is affected by very similar but sometimes very different factors, so it would be great if the analysis can be separated by gender as the sample size allows.

Dear reviewer,

Thank you for the clarification of your previous comment. Now, we understand your point.

However, separating the analysis by gender is beyond the scope of the current paper. The aim of the current paper is to conduct an overview and an exploration of the overall link between body image and the double burden of nutrition (underweight and obesity) by socio-demography.

We already had planned conducting a follow-up paper that will outline the social and psychological predictors of body image. This planned analyses will be acknowledging all the most important sociodemographic (presenting different genders, different ethnicities etc. separately) and economic - related factors that influence one’s body image that we found in this paper, and other health-related factors published in other related published papers in South Africa. Separating these two papers is important in that, it will allow room to unpack and discuss all these determinants separately without overwhelming the reader by lumping all this information in one paper. After-all, Reviewer 3 had suggested that we reduce the information presented in this papers as it was too much already – and consider presenting it in different manuscripts.

2) In Table 1, CIs for "No schooling" and "Grades 1 to 5 (primary school)" are the same. Please make it sure that they are correct.

With regards to comment #2: The confidence intervals in Table 1 have been corrected, thank you.

Final minor spell check has been done.

Kind regards

..........................................................................